# Exploring the Relationship between School Gardens, Food Literacy and Mental Well-Being in Youth Using Photovoice

**DOI:** 10.3390/nu11061354

**Published:** 2019-06-16

**Authors:** Vanessa Lam, Kathy Romses, Kerry Renwick

**Affiliations:** 1Public Health Program—Child and Youth, Vancouver Coastal Health, Pacific Spirit Community Health Centre, 2110 W. 43rd Avenue, Vancouver, BC V6M 2E1, Canada; kathy.romses@vch.ca; 2Department of Curriculum and Pedagogy, University of British Columbia, 2125 Main Mall, Vancouver, BC V6T1Z4, Canada; kerry.renwick@ubc.ca

**Keywords:** mental health, food literacy, photovoice

## Abstract

The goal of the project was to gain an understanding of the relationships between secondary school youth experiences in school gardens and their mental well-being. Over the course of five months, sixteen youths participated in a photovoice research project in which they expressed their personal experiences about food and gardening through photography and writing. The aspects of secondary school youths’ life experiences affected by exposure to school gardens and their impact upon their well-being were identified. The youth explicitly associated relaxation with the themes of love and connectedness, growing food, garden as a place, cooking, and food choices. They were able to demonstrate and develop food literacy competency because of their engagement with the gardening and cooking activities. Youth clubs or groups were identified as a key enabler for connection with other youth and adults. Youth shared their food literacy experiences, observing that their engagement improved some aspect of their mental well-being. Through the photovoice process, the youth identified how their involvement in green spaces enabled connections with others, and highlighted aspects of personal health and personal growth, all of which contribute to their mental well-being.

## 1. Introduction

What many young people might have to say about their health and what they might want or prefer is an important perspective in creating spaces where health and well-being are optimized. This paper reports on a study that focused on youth exploring three interrelated aspects—consideration of mental health and food literacy while situated within a school garden context. The study had two objectives: (1) to identify what aspects of a young person’s life experiences are affected by exposure to food literacy and school gardens; and (2) to identify the components and impact of food literacy and school garden experiences that contribute to young people’s well-being. The opportunity offered by school gardens as green spaces is due to them being widely placed and accessible for youth, particularly in the Vancouver city area. These green spaces enabled experiential, problem-based education while also allowing for interactions and social engagement amongst the “gardeners” [1]. Thus, engaging with green spaces supports positive mental health in youth by building their resilience, sense of autonomy and positive outlook [2,3] as well as a sense of inclusion and belonging. The complimentary focus on food literacy was possible due to a number of the green spaces being school gardens. The youth were able to plant, cultivate, harvest and prepare foods that they had grown. Food literacy is an emerging concept that requires a holistic approach where understandings about food that are socially constructed and able to be shared [4,5,6].

Hands-on food education provides empowerment and skill building that can bolster resiliency to support the mental well-being of students [7] and support academic development [8,9]. Local school and community partners, who have worked with the authors, shared observations and comments related to school connectedness, improved self-esteem, feeling safe in school gardens and mental well-being about students who were involved in school garden and food literacy activities. While similar observations have been noted in other locations including the USA [1], New Zealand [10], Australia [11] and Greece [11] no local research data was available, particularly from a youth perspective.

In this study, photography was chosen as way to enable participating youth to investigate their world, [12,13,14] looking for connections between school gardens and how they perceived their own mental health [12]. Wang [12] describes photovoice as a participatory action research methodology where community members document their everyday realities in photographs, and use critical dialogue to express personal or community strengths and concerns. The use of an arts-based methodology for this study, specifically photovoice, was predicated on the understanding that knowledge exists between and among youth in social settings and that learning is the result of their interactions [15,16]. Thus, the use of photovoice enabled the youth to become empowered through active involvement as co-researchers and engaging with the process of developing knowledge [17].

## 2. Materials and Methods 

### 2.1. Photovoice

Photovoice methodology was developed by Wang and Burris [13] with the specific intent to share expertise and knowledge in participatory ways and to enable people to document community strengths and concerns [11,13]. Using photovoice voice as a method enabled the investigators to work with youth to document their experiences in the garden. The youth took photographs to record aspects of their engagement in the school gardens recording those aspects that took their attention. Follow up conversations and discussions enabled time for reflection about the stimulus for the image [12,13,14]. As a community-based participatory research method, photovoice privileged the voices of youth as they represented their experiences of the school garden green space [15]. The resultant conversations and discussions were possible because of the shared experiences, respectful relationships and level of trust [15,18].

The research team funded a professional photographer, with experience in photovoice projects, to collaborate with the primary investigator to guide planning and facilitation of youth sessions around photography and writing. The photographer helped youth build skills in storytelling through photography, showing examples of other photovoice projects, providing digital camera equipment, training youth on how to operate the digital camera equipment, and collecting and organizing their digital photographs. This process of working with youth to build their photography skills and develop the technical capacity to present their views as both “competent citizens and active participants” [12] (p. 152) who are engaging in transformational acts.

### 2.2. Ethics and Participants

Ethics approval for this study was obtained from The University of British Columbia (UBC) [H15-03101]. Permission to approach and engage with schools in Vancouver was obtained from the Vancouver Board of Education. School and community partners in Vancouver recommended recruiting participants from existing youth clubs and groups. Flyers were distributed in January to community partners and posted at secondary schools to help with recruitment and a Youth Photovoice Project Facebook page was created to share information and updates about the study. Purposive sampling was used to recruit secondary school students from these sources, which provided a diverse group of youth. 

Sixteen youth, nine females and seven males, in grades nine to twelve, attended seven, one to one and a half hour photography and writing sessions over a five-month period. All participants were actively engaged in garden to table activities such as growing, harvesting and cooking. Based on youth preferences, the sessions were held primarily on a weekday after school at their own school site or where their existing club or group was already convening. These sites included two secondary schools, a university farm site, a community center and a neighborhood house. The school clubs and groups were intergenerational social contexts in that there were adults who worked as facilitators. These adults interacted with youth in the gardens and in doing so connections and relationships were developed.

### 2.3. Project Steps

The youth joined in the three stages of the participatory approach of photovoice methodology described by Wang and Burris [13]. The first stage is having youth take their photographs and selecting their photographs of choice. The second stage focused on contextualizing or providing stories about the meaning behind the chosen photos. The sessions with youth included having them document their reflections in personal journals and sharing some of their writing with investigators and fellow youth. The third stage involved codifying or identifying themes from the research process, which took place both individually with each of the investigators and then together, followed by identifying and checking themes with youth co-researchers. Two focus groups were facilitated one in June and the second in September.

Since different methods were used to collect information to triangulate findings and increase rigor and credibility a meta-analysis was facilitated. Each investigator used a holistic thematic approach by independently reading the transcribed interviews, focus group transcript and youth writing. One investigator utilized the *NVivo 10.0* [19] computer software and the other investigator used a manual review process to develop a preliminary coding framework that emerged from the data. They met several times to iteratively develop a shared list of categories, overarching themes and sub-themes to enhance inter-rater reliability.

Based on the derived draft list of themes and categories, overarching themes were shared for participant verification at the September focus group to further improve inter-rater reliability and verify findings with youth co-researchers. Five females and one male were in attendance. Two male participants joined through a video Skype call on a mobile phone. A mind mapping activity was utilized to allow youth to further refine and validate the key themes and connections between themes.

## 3. Results

The data emerged out of this study in a number of different ways—visual representations in the photographs, the associated text and ongoing conversations. As the youth were setting up and reflecting on their photographs and text, they were asked brainstorming questions as a way to help youth guide their photography and story writing. The examples of questions included: “What does mental well-being mean to you? and What does food and gardening mean to you?” After the group discussions, personal diarizing and selection of images the youth and investigators were able to identify three broad emergent themes—connection, personal growth and personal health as being important to mental well-being. The engagement in the photovoice process around the experiences of food, gardening and well-being offers insights into what the youth were able to identify as strengths and concerns at both a personal and community level.

### 3.1. Text and Images

Participants indicated that connecting comes from having community, relationships and respect. Youth clubs or groups were identified as a key enabler for connection. The use of a garden offered opportunity to be with others that was different to their experience in school classrooms. 

The image and text provided in Figure 1 presents one young person’s take on their involvement in the Carrot Club. This was an after school program that cooked foods including those harvested from the school garden. The club leaders structured the various activities in ways that facilitated interactions between the youth who attended.

Table 1 offers examples of what youth perceived as being about connections that emerged in the conversations and discussions. The idea of connections was about community and could be seen in their relationships, how they work in respectful ways. The intergenerational aspect of gardens was seen in a positive way, as was the capacity of the garden context to accommodate differing levels of expertise. The perceptions about the inclusive and non-judgmental aspects of the garden offered spaces where the youth could see themselves as having a legitimate place with others, an opportunity to contribute and achieve. 

Being in gardens and the growing of food facilitated reflections about time, continuity and change. Engaging with the photovoice process enabled this participant to imagine how an image of a lone strawberry seen in Figure 2, could equate to people being on their own, taking time for self-discovery. The image and text tells a story about an understanding that a youth photographer has and wants to share [12].

Table 2 provides examples of what youth described as personal growth through self-discovery and across time. Participants identified those things that “change all the time”, such as self-discovery, life, personal growth, slower pace, future and nurture. Time spent in the garden enabled a different engagement with the world. That instead of seeing it through computer or phone screens, these could be set aside to experience what is necessary to grow something and that this could inform how they lived their own lives. 

Within the theme of personal health, identified factors included emotions and feelings, having a safe place, nutrition and relaxation. Relaxation was connected with the following grouped themes of: growing food, garden as a place, cooking, love and food choices. This was linked to nature, beauty, environment and sustainability. In the photovoice contribution shown in Figure 3, the participant uses prose to help to explain the image offering some insight into the participant’s interests [12]. The image presents hands cradling a young plant with roots embedded in soil. It evokes a sense of gentle care with the text highlighting the resultant potential.

The statements in Table 3 make links between personal health and nature, beauty, environment and sustainability. The participants acknowledged how being in the garden with others impacted on their personal health. Their considerations included how consumption of food nurtured their body in a physical sense but also how it enabled social health through engagement and shared activity with others. That well-being was observed in the participants’ positive sense of self and environmental awareness.

The use of photovoice as method allowed the youth as participants to document their experience of being in the garden with others, thinking about the image, its composition, use of light and texture, shape and form all come together to convey the youth’s impressions about food, gardening and mental well-being. The accompanying text highlights a particular experience that the young person wanted to convey about themselves and their relationships with others. Evident in these artifacts is a sense of agency [20] and personal growth due to the interactions both with the school garden as a green space and with other gardeners [18].

### 3.2. Evidence of Food Literacy

Food literacy is a sociocultural knowing where food-related activities are based on particular understandings that are context dependent and infused with power-relations [4,5]. The participants’ in this study were not directly engaged with considerations and reflections on food literacy *per se*. Rather the researchers focused on statements made in the conversations and discussion with the participants. Thus the researchers were exploring food literacy based on the exploration of school gardens by the youth. Categorization of these statements was informed by an understanding of food literacy utilizing a multidimensional model developed by Green [21] and adapted for food literacy by Renwick [5]. This model describes three different but related dimensions—operational, cultural and critical, and was used as a way to elicit the researchers’ understanding about the ways the youth as participants were developing food literacy.

Operational food literacy is about engaging with practical aspects of working with food. It is seen in the ways food is selected and prepared, how it is stored and preserved and where it comes from. For example, when youth spoke about their experiences of planting, growing, harvesting and preparing food, these were aspects of the operational dimension of food literacy. Such comments included:
For me [I] like food workshops, we make foods; I like to make pizza, or pretzels. Pretzel is fun.
It’s really about the skills, something you gonna do when you grow up.
Between cooking and gardening we use cooking more in the future because we have to eat.

Cultural food literacy refers to how food is used in cultural contexts to convey social meaning. Youth spoke about their food experiences and how they connected to family as a significant space for induction into culture. They drew on ways of knowing, especially around traditional and healing plants, reflecting a cultural dimension of food literacy. Their cultural food literacy included the retelling of experiences with family members, intergenerational learning and connecting to others with expertise.
A lot of the stuff [my grandfather] showed me are pretty simple things that gave me a little head start with teaching the kids… how to plant things, like how far do they have to be, like the spacing. Making sure that if they have to dig something up, if there’s a rock or something, try to find a way around it, not just dig the rock out. And leave the rock somewhere else far away from the plant.
It’s easier in the garden to talk about and learn about other people.
I always go with my mom to the market, like supermarket or market place, buy vegetables and rice, so I know how to pick them. Like choosing apples or oranges as fruits, or cabbage something like that.

Critical food literacy is implicated in our social practices and exercising of power relations because of how we act as eaters and consumers. The critical dimension was evident when the youth discussed how they connected the growing of food with eating healthy or sustainable food. When engaging with this dimension youth were able to connect their experiences to what is occurring elsewhere. For instance, their efforts to produce food was scaled up in their imaging of what those who work on the land would need to do to feed communities.
Although it’s fun and rewarding, it’s a lot of work to grow something and when it dies you’re kind of a little bit sad but it’s part of it I guess and you kind of realize how much work it is, like to grow one plant, it took us a lot of work and then I think about the farmers that do like that for a living and then I’m like, you kind of realize how hard they work and you appreciate it more.

Drawing on Cullen et al.’s [22] definition of food literacy, these quotes highlight what youth are thinking about their food experiences demonstrated an understanding of what is involved in growing and preparing food. They also highlighted the relationship between food and the youth, as evidenced in ‘real’ social contexts, which supported their connectedness with others. Thus food literacy is concerned with the interactions and concepts about food, typified as farm to table and how it has meaning in daily life [22,23]. The very processes of selecting, producing and preparing food for consumption are implicated with physical, emotional and mental health.

## 4. Discussion

This paper contributes to the growing research that focuses on the relationship between school gardens and student mental health. The research provided an in-depth understanding of the intersection of mental health and food literacy from a youth’s perspective. All of the youth shared that their food literacy experiences improved some aspect of their mental well-being that included building their confidence, resiliency and sense of self-mastery. Seventy-five percent of Vancouver public schools have food gardens, which can provide opportunities to be outside and be involved with hands-on, interactive learning. Fostering opportunities for food literacy such as growing and preparing food could help build resiliency. These strategies can build a supportive environment around mental health promotion in schools [24]. The school garden can be used to promote physical activity, mental health and well-being, neighborhood connections [1,14], as well as cross-curricular and inquiry-based learning and educational outcomes [1,25]. The attention restoration theory explains the link between spending time in nature such as the school garden with being able to concentrate better [26]. This is supported by the qualitative research by Chawla et al. [25], which involved youth in gardening programs. The use of the photovoice method enabled youth to share their understandings of way they considered as their personal strengths and concerns for their mental well-being while engaged in school gardens growing food.

The Comprehensive School Health (CSH) framework is recognized internationally and is used by public health practitioners and school stakeholders in Vancouver. The framework helps to build sustainable changes in school culture that support the health, learning and well-being of students. Storey et al. [27] identified core conditions for CSH implementation that include students as change agents, school-specific autonomy, demonstrated administrative leadership, dedicated champion to engage school staff, community support, evidence and professional development. The contextual conditions of time, funding and project supports, readiness and prior community connectivity influence the capability of the core conditions to be achieved. The current study addressed the core conditions of secondary school students as change agents and generated evidence from the local school district setting. This supports Wang and her colleagues’ [12,13,14] recommendations for action based on findings from photovoice projects. The action part of the process for the present study is ongoing and has involved two public art gallery exhibitions; mobile displays at school board lobby, a secondary and elementary school; sharing research summaries with the funding group and fellow researchers, school board Research Committee, school board Sustainability Manager and Coordinator; an oral presentation at a university dietetic intern research symposium; a poster presentation at a regional health authority workshop for dietitians and part of a walking tour at a national Dietitians of Canada conference.

The use of a research method with a qualitative focus and the resultant small sample size meant that the specific findings are not intended to be generalized beyond the study participants and localized context. However, the general findings may be applicable to other secondary school youth involved in food literacy, in particular school gardens and related school and community activities. Further research on the long-term impact of this photovoice research project on food- and garden-related school and community activities is warranted.

The intention of this project was to engage with youth as co-researchers to explore the relationship between their well-being and green spaces such as school gardens. Thus, there was an exploration of engaging with green spaces that both supported mental health but also supported the development of food literacy. Based on this project, the researchers observe that food literacy in a cultural context such as growing and preparing food can build resiliency in youth as a promising area for further work. Additionally this experience supports how incorporating the perspective of youth can impact their sense of connectedness and engagement with learning.

The use of the photovoice method enabled youth to share their understandings of way they considered as their personal strengths and concerns for their mental well-being while engaged in school gardens growing food. In responding to the two objectives of this study it was noted that there were a number of aspects of a young person’s life experiences that are affected by exposure to food literacy and school gardens. The youth participants presented a number of understandings about each of the three dimensions of food literacy drawing on specific personal experiences. While not an explicit focus of the work with youth they were able to provide examples reflecting on experiences in food handling, growing particular foods, to thinking about their personal and family food preferences, to their role in the food system. And secondly youth did not identify the specific components of food literacy referred to in this study. However, they were able to describe ways in which their growing food literacy linked in with their school garden experiences and contributed to their mental well-being.

## Figures and Tables

**Figure 1 nutrients-11-01354-f001:**
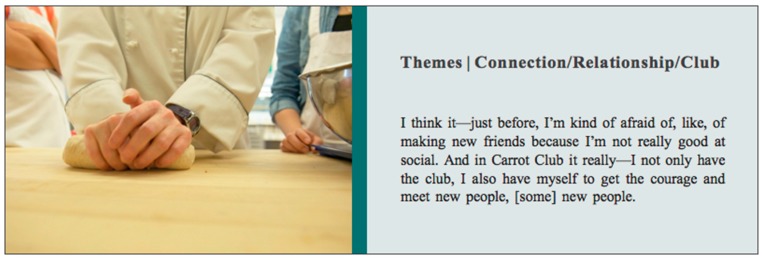
Photovoice artifact.

**Figure 2 nutrients-11-01354-f002:**
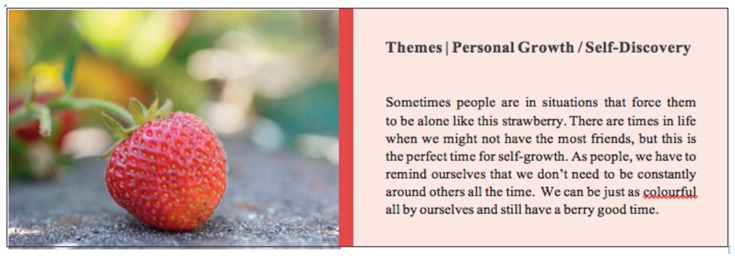
Photovoice artifact.

**Figure 3 nutrients-11-01354-f003:**
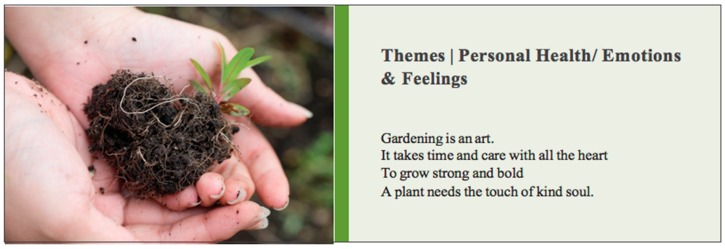
Photovoice artifact.

**Table 1 nutrients-11-01354-t001:** Theme: connections.

Sub-Theme	Sample Participant Responses
Relationship	Well, the most I can say is I’ve never seen—well, like, if I have, it’s very rare, but I’ve never seen, like, both youth and seniors working together, and it’s gardening. And like, because usually when it’s—when I think of teenagers, I think of people, like, using their phones every five minutes, and this is not something I saw during the times we were gardening. And so yeah, that’s just something I noticed. And it was very interesting, and you need to see something like that, to see it, that gardening can do this to people.
Community	I’ve seen it in gardening mostly, that people do come together of all ages. Like, you don’t need a lot of experience or not a lot of knowledge to actually be able to do gardening. Like, you can just pick up and do something in the smallest way and it can help. And you see, like, the actual growth and the actual outcome of your own efforts, so it’s really rewarding.
	Well, I see people laughing, making jokes about small things, and just enjoying the process of gardening. It’s not—it’s not really actual gardening, but it’s working with other people, other peers. And—yeah.
Respect	I think that the biggest thing is that when you’re gardening, that there’s kind of like a sense of, like, no judgment of what you’re doing in the moment. So you can share your opinions and really get to—get to know people from, like, a different point of view, and get to see, like, what their skills that they’re good at. So in terms of mental wellbeing, I would say it really makes you find that connection of what you’re good at, and makes you feel good about yourself in the end.

**Table 2 nutrients-11-01354-t002:** Theme: personal growth.

Sub-Theme	Sample Participant Responses
Self-discovery	You don’t always have to be around others, and especially for youth, at my age or—we kind of feel the need to fit in and always be around people, like social media and things like that. But I think what’s more important is, like, finding that balance in order to solve that problem, and finding that balance might mean something like gardening or, like, doing something else you love.
	Well, especially this age, it’s just, it comes from peers, like, kind of that pressure to always fitting—to always fit in and, like, always be talking to other people, but sometimes we have to take that time off for ourselves, for, like, self-growth, and how we can still be bright and colorful just on our own. Well, it’s again, like, taking time, doing things like gardening and like, finding that balance in our lives, just focusing on multiple things, not just one, in order to—in order for that self-growth to happen.
	how gardening, it—there’s, like—it can relate to everyday life, not—it’s not just gardening.
Future	I would say give the opportunity to students to actually bring out what they think is important, and to not only teach what is part of the curriculum, but something that could help nurture what they want to do in life.

**Table 3 nutrients-11-01354-t003:** Theme: personal health.

Sub-Theme	Sample Participant Responses
Nutrition	When I started working in the garden, I started to feel this great connection with the things. I grew, and I started to care more about what I was eating.
	When you’re the one who grew the food, like, it just adds that extra sort of excitement and makes you want to eat the food.
	We heal ourselves using the foods we eat. So that’s part of the problem I see, is that people don’t care enough about the—about themselves and their health and the environment, the planet, because that’s all connected.
Relaxation	It’s therapeutic in some way actually. Like, touching the soil is actually calm—not calm—like, calming, but, like, it’s nice to just garden and get your hands dirty and grow stuff and seeing it grow, watching it grow.
	[Gardening] gave me more friends I guess and it’s kind of like what I said last time is [it’s] like a safe place where I can escape from problems or stress that I have during school and instead I can embrace the things that make me happy.
	It calms me and allows me to think, not negatively, but logically.

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
