# Peer review of "Exploring the Relationship between School Gardens, Food Literacy and Mental Well-Being in Youth Using Photovoice"

_nutrients, 2019, doi:10.3390/nu11061354_

Round 1

Reviewer 1 Report

I love the program and the implications it has on the mental and physical well-being of not only our youth but our communities. The garden is a place of healing and this is not just anecdotal. It decreases feelings of anxiety and depression as well as increases feelings of belonging.

Author Response

Thank you for your response.

We have edited the paper and hopefully addressed all of the required aspects.

Reviewer 2 Report

See attached.

Author Response

Thank you for your response.

We have edited the paper and hopefully addressed all of the required aspects.

Please see attached document for specific details

Round 2

Reviewer 2 Report

The authors made the requested changes and have much improved the manuscript. The implications of the study could be made clearer and placed within a stronger framework--hence the "low" score for significance of content--but overall it's a sound report that may be of interest to some readers.